# Effects of Different Forms and Proportions of Nitrogen on the Growth, Photosynthetic Characteristics, and Carbon and Nitrogen Metabolism in Tomato

**DOI:** 10.3390/plants12244175

**Published:** 2023-12-15

**Authors:** Jianhong Sun, Li Jin, Ruirui Li, Xin Meng, Ning Jin, Shuya Wang, Zhiqi Xu, Zitong Liu, Jian Lyu, Jinhua Yu

**Affiliations:** 1College of Horticulture, Gansu Agricultural University, Lanzhou 730070, China; 13993965642@163.com (J.S.); 18409318527@139.com (R.L.); mx17393157902@163.com (X.M.); jinn0513@163.com (N.J.); 15294198598@163.com (Z.X.); liuzitong2000@126.com (Z.L.); 2State Key Laboratory of Aridland Crop Science, Gansu Agricultural University, Lanzhou 730070, China; jinli0124@163.com (L.J.); wsyhn95@163.com (S.W.)

**Keywords:** tomato, nitrogen form, nitrogen metabolism, Calvin cycle, sugar metabolism, photosynthetic characteristics

## Abstract

Optimal plant growth in many species is achieved when the two major forms of N are supplied at a particular ratio. This study investigated optimal nitrogen forms and ratios for tomato growth using the ‘Jingfan 502’ tomato variety. Thirteen treatments were applied with varying proportions of nitrate nitrogen (NN), ammonium nitrogen (AN), and urea nitrogen (UN). Results revealed that the combination of AN and UN inhibited tomato growth and photosynthetic capacity. Conversely, the joint application of NN and UN or NN and AN led to a significant enhancement in tomato plant growth. Notably, the T12 (75%UN:25%NN) and T4 (75%NN:25%AN) treatments significantly increased the gas exchange and chlorophyll fluorescence parameters, thereby promoting the accumulation of photosynthetic products. The contents of fructose, glucose, and sucrose were significantly increased by 121.07%, 206.26%, and 94.64% and by 104.39%, 156.42%, and 61.40%, respectively, compared with those in the control. Additionally, AN favored starch accumulation, while NN and UN favored fructose, sucrose, and glucose accumulation. Gene expression related to nitrogen and sugar metabolism increased significantly in T12 and T4, with T12 showing greater upregulation. Key enzyme activity in metabolism also increased notably. In summary, T12 enhanced tomato growth by upregulating gene expression, increasing enzyme activity, and boosting photosynthesis and sugar accumulation. Growers should consider using NN and UN to reduce AN application in tomato fertilization.

## 1. Introduction

Nitrogen is an essential plant nutrient, and inhibition of its supply can reduce crop growth and yield [1,2]. Plants acquire nitrogen in various forms from the soil through processes like transport, assimilation, and remobilization [3,4]. The primary nitrogen forms are ammonium nitrogen (NH_4_^+^), nitrate nitrogen (NO_3_^−^), and amide nitrogen (CO(NH_2_)_2_) [5], with NH_4_^+^ and NO_3_^−^ being the key forms taken up by plants [6] through specific transport mechanisms [7]. Urea, known for its stable chemical properties and high nitrogen content, is commonly used in agriculture. Soil-bacteria-driven urease degradation converts urea into NH_4_^+^, which is then transported to plants, while unaltered urea is also transported to plants via high-affinity transport systems [8]. Plant nitrogen absorption and distribution are influenced by factors such as species variation, growth stage, nitrogen assimilation enzyme activity, and nitrogen form. A study using lettuce showed that it grew well when nitrate nitrogen was used as the nitrogen source. However, when the proportion of ammonium nitrogen in a nitrate and ammonium mixture reached 50%, lettuce growth was found to be inhibited to a certain extent, and consequently, lettuce growth is challenging when the only nitrogen source is NH_4_^+^-N [9]. Wheat, under different nitrogen conditions, exhibits varying nitrogen absorption and utilization efficiencies. Combining urea and nitrate nitrogen significantly boosts wheat yields, while ammonium and nitrate nitrogen enhance nitrogen metabolism enzyme activity during early grain filling stages, improving nitrogen utilization efficiency [10]. For rice, ammonium nitrogen is the preferred nitrogen source, while tomatoes prefer nitrate [11].

Many studies have shown that nitrogen form can affect chlorophyll synthesis and the activity of photosynthesis-related enzymes, thereby directly or indirectly participating in the regulation of plant photosynthesis. Studies have also shown that the photosynthetic rate of plants supplied with NH_4_^+^ is higher than those provided with NO_3_^−^ [12]. However, other studies have shown that high concentrations of nitrate nitrogen can reduce the photosynthetic rate of plants [13]. Raab et al. [14] found that the photosynthetic efficiency of sugar beet leaves supplied with NH_4_^+^ was lower than that of those supplied with NO_3_^−^. The mixed application of NH_4_^+^-N and NO_3_^−^-N increased the net photosynthetic rate (Pn), stomatal conductance (Gs), transpiration rate (Tr), intercellular CO_2_ concentration (Ci), and other gas exchange parameters when compared with a single nitrogen source [15]. Nitrogen also affects the activity of the photosynthetic carbon assimilation enzymes. The activity of RuBPcase decreases under the action of a high concentration of NH_4_^+^ [16], and this may be due to the toxic effects of NH_4_^+^. Excessively absorbed NH_4_^+^ undergoes conversion into ammonia during membrane transport, primarily in the aboveground plant parts. Ammonia causes plasma membrane depolarization, disrupting photosynthetic phosphorylation and hindering CO_2_ fixation, ultimately leading to reduced photosynthetic efficiency.

Research into the influence of nitrogen levels on sugar metabolic enzyme activity has primarily centered on sucrose metabolism. Sucrose phosphate synthase (SPS) and sucrose synthase (SS) represent pivotal enzymes in the regulation of sucrose metabolism. An optimal nitrogen application level can enhance the activity of SPS and SS enzymes in the leaves. Conversely, inadequate or excessive nitrogen application results in decreased enzyme activity [17,18]. Research has explored the impact of nitrogen deficiency on carbon assimilation in plants that produce sucrose and starch as end products of photosynthesis. Studies on fruit trees have indicated that nitrogen deficiency can result in a reduced capacity for CO_2_ assimilation. This reduction is primarily attributed to non-stomatal factors, as the levels of intercellular carbon dioxide in apple leaves remain elevated in the absence of nitrogen [19]. The synthesis and accumulation of carbohydrates in plants are initially achieved through photosynthesis, which relies on numerous proteins and enzymes within the photosynthetic system. Nitrogen deficiency can significantly diminish the activity of RuBP carboxylase/oxygenase (Rubisco) and other enzymes involved in photosynthesis [20]. Furthermore, it can alter the distribution of assimilation among different plant organs. In the lag phase of grain filling, a low nitrogen supply promotes the distribution of the latest assimilated photosynthetic products to the stems and roots and reduces their distribution to the reproductive organs. Although photosynthetic assimilates transported to the reproductive organs decrease, the concentration of sugar in the seeds increases [20]. Chen et al. [19] observed that reduced nitrogen application levels led to a decrease or no significant change in the levels of sucrose, fructose, and glucose in the leaves of ‘Gala’ apple trees. Interestingly, nitrogen deficiency was found to have a direct impact on the activity of crucial enzymes involved in the Calvin cycle and the synthesis of photosynthetic products. This effect was attributed to feedback inhibition related to the accumulation of soluble carbohydrates rather than carbon assimilation.

Tomato, being a vital vegetable crop with substantial economic and nutritional value, is known to rely heavily on nitrogen for its growth and development. It is crucial to maintain an appropriate nitrogen supply, as both excess and insufficient nitrogen can impose limitations on the growth and development of tomato plants [21]. Therefore, when providing a specific quantity of nitrogen, the nitrogen form becomes a critical factor influencing plant growth and development. The application method of nitrogen has long been a subject of investigation. While previous research has explored the mixed application of ammonium and nitrate nitrogen, there has been limited investigation into the role of amide nitrogen in tomato photosynthesis and its associated internal enzymes and molecular mechanisms. To delve deeper into the potential mechanisms through which different nitrogen forms in nutrient solutions can enhance plant photosynthesis, further research is needed to validate how various nitrogen forms can promote plant photosynthesis. In light of these considerations, gaining insights into tomato’s preference for different nitrogen fertilizer forms is essential when formulating effective fertilization strategies.

In this experiment, potted tomato plants served as the subjects, and various combinations of three distinct nitrogen forms (ammonium nitrogen, nitrate nitrogen, and amide nitrogen) were applied. This study encompassed an examination of gas exchange parameters, chlorophyll fluorescence imaging, the activities of pivotal enzymes associated with the Calvin cycle, carbon and nitrogen metabolism, the relative expression levels of genes encoding these enzymes, and the accumulation of photosynthetic products (total soluble sugar and starch) within tomato leaves. The primary objective was to pinpoint the optimal nitrogen form ratio for enhancing tomato growth while elucidating the regulatory mechanisms by which these diverse nitrogen forms influence tomato development. This research was intended to furnish technical guidance and a theoretical foundation for the effective regulation of tomato growth.

## 2. Materials and Methods

### 2.1. Plant Materials and Growth Conditions

This experiment took place within a glass greenhouse situated at coordinates 36°05′39.86″ N and 103°42′31.09″ E, located in the College of Horticulture of Gansu Agricultural University in China. The test material employed for this study was the tomato variety *Solanum lycopersicum* cv. Jinfan502. The tomato seeds underwent a series of preparatory steps: they were initially soaked in warm water with temperatures ranging from 55 to 60 °C, stirred for 30 min, and subsequently immersed in water at 28 °C for 8 h. Afterward, these seeds were evenly distributed on a culture plate containing moist filter paper and placed in a dark climate-controlled chamber set at 28 °C with 75% humidity, allowing for germination over a span of 30 h. Once the germination rate reached 80%, the seeds were transplanted into a 50-hole plug tray filled with a seedling substrate. These trays were then positioned within a climate-controlled chamber featuring 12 h of daylight at a photon flux density of 320 μmol·m^−2^·s^−1^. The conditions inside the chamber were set at 28/20 °C (light/dark) and maintained at a relative humidity of 75% to facilitate seedling growth. Subsequently, when the seedlings developed 4–5 leaves, they were transplanted into flowerpots with dimensions of 30 cm in diameter and 20 cm in height. These flowerpots were filled with a cultivation medium composed of Lv neng rui qi substrate, peat, and vermiculite in a ratio of 2:1:1. The plants were then relocated to a greenhouse environment with temperatures of 30 ± 2 °C/18 ± 2 °C (day/night), with a photoperiod of 12 h/12 h (day/night), and the relative humidity was maintained at 60–70%.

### 2.2. Experimental Design

In this study, various nitrogen sources were employed as the primary nitrogen input. After one week of transplanting the seedlings, a nutrient solution was applied every 3 days. Specifically, each pot received 1000 mL of a nutrient solution prepared based on the Hoagland formula. However, to create distinct nutrient solution formulations with varying nitrogen forms, adjustments were made to the nitrogen component. These custom formulations were labeled as different treatments. Throughout the experiment, all treatments maintained consistent concentrations of N (15 mM), P (1 mM), K (6 mM), Mg (2 mM), and Ca (5 mM). The key difference lay in the ratios of nitrogen forms used, denoted as T1–T12, while a control group with no nitrogen fertilizer was designated as CK. Nitrate nitrogen was provided by KNO_3_ and Ca(NO_3_)_2_, purchased from Sinopharm Chemical Reagent Co., Ltd. (Shanghai, China). Ammonium nitrogen was provided by (NH_4_)_2_SO_4_, purchased from Tianjin Zhiyuan Chemical Reagent Co., Ltd. (Tianjin, China). Amide nitrogen was provided by urea, purchased from Anyang Zhongying Chemical Fertilizer Co., Ltd. (Anyang, China). The magnesium source was MgSO_4_·7H_2_O; the phosphorous source was KH_2_PO_4_; the potassium sources were KCl, KNO_3_, and KH_2_PO_4_; the calcium sources were CaCl_2_ and Ca(NO_3_)_2_, and trace elements were configured according to the Hoagland nutrient solution. To prevent the conversion of ammonium ions into nitrate ions in the nutrient solution, the nitrification inhibitor dicyandiamide (C_2_H_4_N_4_, 7 μmol·L^−1^) must be added to the nutrient solution. The ratio of N, P_2_O_5_, K_2_O, CaO, and MgO required for tomatoes is approximately 1:0.29:1.6:1.2:0.26; according to the total nitrogen application rate of 300 kg·hm^−2^, the transplanting density was 37,500 plants per hectare, converted to the amount of fertilizer per plant for fertilization, and the average application of pure N, P, K, Ca, and Mg per plant is approximately 8.00 g, 2.32 g, 12.80 g, 9.60 g, and 2.08 g. The experimental settings are listed in Table 1.

### 2.3. Determination of the Indexes and Methods

#### 2.3.1. Determination of the Tomato Growth Index

For each treatment, we carefully selected and labeled tomato plants with uniform growth characteristics (5 plants per treatment). Subsequently, we monitored their growth indices, including plant height, stem thickness, and leaf area, at 15-day intervals. Plant height was measured from the root to the highest point of vertical growth using a measuring tape. The diameter of the plant, located 4 cm above the base, was measured using an electronic Vernier caliper from Shanghai Mediante Industrial Co., Ltd. (Shanghai, China). For leaf measurements, we focused on the fifth leaf of each tomato plant. We measured its length (L) and width (W) with a ruler and then calculated the leaf area using the following formula [22]:The leaf area of leaflets (leaf length ≤ 30 cm) LA = 0.228LW + 8.152 (1)
The leaf area of the large leaf (leaf length > 30 cm) LA = 0.233LW + 31.387 (2)


L: The distance from the petiole base to the leaf tip;W: The maximum width perpendicular to the main vein.


#### 2.3.2. Determination of Root Morphological Parameters

On the 30th day of fertilization, three tomato plants with uniform growth were selected from each treatment and cut from the base of the stem, and the roots were carefully washed with distilled water. Tomato roots from each treatment were scanned using an EPSON expression 11000XL scanner (Win RHIZO Pro LA2400, Regent Instruments Inc., Quebec City, QC, Canada), and scanned photos were analyzed using the Win RHIZO 5.0 software (Regent Instruments Inc.) to obtain the total root length, root surface area, root volume, average root diameter, and the number of root tips and forks. Root activity was determined using the triphenyl tetrazolium chloride (TTC) method [23].

#### 2.3.3. Determination of Gas Exchange Parameters

On the 60th day following fertilization, we carefully selected three tomato plants from each treatment group with similar growth characteristics. Subsequently, we measured the gas exchange parameters, including Pn, Gs, Tr, and Ci, of the third functional leaf, which had a consistent size and was located beneath the growth point. These measurements were conducted using a CIRAS-2 portable photosynthesis system from the British PP System Company. The instrument’s settings were as follows: a temperature of 25 °C, a light intensity of 1000 μmol·m^−2^·s^−1^, ambient CO_2_ concentration, and a relative humidity of 75% [24].

#### 2.3.4. Determination of Chlorophyll Fluorescence Parameters

On the 60th day after fertilization, we chose three tomato plants from each treatment group with consistent growth patterns. These selected plants were kept in darkness for 30 min. Afterward, we picked functional leaves of the same size and location to assess chlorophyll fluorescence parameters. To perform these measurements, we utilized a modulated chlorophyll fluorescence imager (Walz; Effeltrich, Germany) [15,25,26]. The intensity of the detection light was set to 0.1 μmol·m^−2^·s^−1^, the intensity of the photochemical light was set to 111 μmol·m^−2^·s^−1^, the intensity of the saturated pulse light was set to 2700 μmol·m^−2^·s^−1^, pulse light saturation time was 0.8 s, and time interval was 20 s.

#### 2.3.5. Determination of Photosynthate Content

On the 60th day after fertilization, we selected three tomato plants with consistent growth for each treatment. We specifically chose leaves that were of the same size and in a similar position on the plant to assess the content of photosynthetic products. We followed a slightly modified [27] version of the method described by Jin Ning and others to determine the glucose, fructose, and sucrose contents in these leaves using high-performance liquid chromatography. The process involved grinding 5 g of leaves, transferring them to a 25 mL volumetric flask, and adjusting the volume with ultrapure water. Next, the mixture was placed in a water bath and subjected to ultrasonication at 30 °C for 60 min. Afterward, it was filtered into a 50 mL centrifuge tube and centrifuged at 4 °C and 10,000 r·min^−1^ for 10 min. The 2 mL supernatant was further filtered using a 0.22 μm water filter, and the resulting filtrate was utilized for the determination of glucose, fructose, and sucrose content. We employed a high-performance liquid chromatograph (HPLC, Agilent 1100, Santa Clara, CA, USA) equipped with a differential refractive index display for this analysis. The chromatographic column used was an LC-NH_2_ amino column (460 mm × 250 mm), with a mobile phase composition of V (acetonitrile):V (water) = 75:25, a flow rate of 1.0 mL·min^−1^, a column temperature of 30 °C, and an injection volume of 20 μL.

Starch content was determined using the iodine chromogenic method [28]. On the 60th day of fertilization, three tomatoes with uniform growth were selected from each treatment, and functional leaves of the same size and position were selected for starch content determination. Tomato leaves (5 mL) were ground with 80% ethanol, and then centrifuged in a centrifuge tube. The residue was washed once with 5 mL distilled water, and then 5 mL 80% Ca(NO_3_)_2_ was added to the residue in boiling water for 10 min. After low-speed centrifugation, the supernatant was transferred to a 20 mL volumetric flask, the extraction was repeated 2 times, combined with the extract, and a constant volume of 20 mL was obtained. Starch samples (0.2 mL) were added to the 2.0 mL with 80% Ca(NO_3_)_2_, and 100 μL 0.01 mol·L^−1^ I_2_-IK was added. The sample was then shaken and the absorbance was measured at 620 nm. The starch content of each sample was calculated by substituting it into a standard curve.

#### 2.3.6. Determination of the Activities of Key Enzymes in the Calvin Cycle, Enzymes Related to Glucose Metabolism, and Enzymes Related to Nitrogen Metabolism

On the 60th day following fertilization, we carefully selected three tomato plants of uniform size from each treatment group. From the top of each plant, we took a third functional leaf of the same size, weighing precisely 0.5 g, and rapidly froze it using liquid nitrogen. An appropriate amount of liquid nitrogen was added to a mortar and the sample was grounded into a fine powder. This powder was then transferred to a 5 mL centrifuge tube. To create a homogenate, we used pre-chilled PBS buffer (pH 7.4) and centrifuged it at 2124× *g* for approximately 20 min at 4 °C. Subsequently, we carefully collected the supernatant for further analysis. The collected supernatant was used to determine the activities of key enzymes involved in various metabolic processes, including the Calvin cycle (such as ribulose-1,5-bisphosphate carboxylase/oxygenase or Rubisco), glucose-metabolism-related enzymes (e.g., glyceraldehyde-3-phosphate dehydrogenase or GAPDH, fructose-1,6-bisphosphate esterase or FBPase, fructose-1,6-bisphosphate aldolase or FBA, transketolase or TK, sucrose synthase or SS, sucrose phosphate synthase or SPS, acid invertase or AI, and neutral invertase or NI), and nitrogen-metabolism-related enzymes (like nitrate reductase or NR, nitrite reductase or NiR, glutamine reductase or GS, glutamate reductase or GOGAT, and glutamate dehydrogenase or GDH).

We used enzyme-linked immunosorbent assay (ELISA) kits (Shanghai Guduo Biotechnology Co., Ltd., Shanghai, China) to determine the activities of these enzymes. Following the manufacturer’s instructions, we added the appropriate reaction solutions. Using a microplate reader (SpectraMax CMax Plus, Molecular Devices, San Jose, CA, USA), we measured the color of the solutions at 450 nm and calculated the enzyme activity for each treatment based on the corresponding standard curve.

#### 2.3.7. RNA Extraction and RT-qPCR Analysis

On the 60th day of fertilization, three tomato plants of uniform size were selected for each treatment and a third functional leaf of the same size at the top was used for RNA extraction and real-time fluorescence quantitative analysis. Each replicate weighed 1 g after grinding in liquid nitrogen, and an RNA extraction kit (Tiangen Biochemical Technology Co., Ltd., Beijing, China) was used, according to the manufacturer’s instructions, to extract total RNA from the tomato leaves. Residual DNA was removed using a gDNA Clean kit (Nanjing Novzan Biotechnology Co., Ltd., Nanjing, China) and the total RNA of the leaves was reverse transcribed. Quantitative reverse transcription polymerase chain reaction (qRT-PCR) experiments were performed using the SYBR Green PreMix Pro Tap HS qPCR kit (Ecoray Biotechnology Co., Ltd., Lanzhou, China). Quantitative analysis was performed using a LightCycle 96 real-time fluorescence quantitative PCR instrument (Roche, Basel, Switzerland). The amplification conditions were 95 °C for 30 s, then 40 cycles of 95 °C for 5 s and 60 °C for 30 s. The tomato actin gene (Actin) was used as the internal reference gene. The primer sequences are shown in Table 2. The relative expression of the genes was calculated using the 2^−ΔΔCt^ method, and each gene expression analysis was performed with three independent biological replicates.

### 2.4. Data Analysis

The data are expressed as the mean ± standard error (SE). One-way analysis of variance was performed using SPSS (version 23.0; SPSS Inc., Chicago, IL, USA). Duncan’s new complex range method was used for the significance test to evaluate the differences between treatments (*p* < 0.05). Excel 2016 and Origin 2022 (Origin Lab Institute Inc., Northampton, MA, USA) were used for the data processing and image generation.

## 3. Results

### 3.1. Effects of Different Forms of Nitrogen on Tomato Plant Growth Parameters 

The various nitrogen treatments had notable impacts on the growth parameters of tomato plants, including plant height, stem thickness, and leaf area, as illustrated in Figure 1. After 60 days of fertilization, all treatments exhibited significant enhancements in plant height (Figure 1A), stem thickness (Figure 1B), and leaf area (Figure 1C) when compared to the control group. Furthermore, during the growth period, these growth parameters experienced rapid increases between days 45 and 60. It is worth noting that different nitrogen forms exerted distinct effects on the growth and development of tomato plants. Notably, the treatment with a nitrogen ratio of 25% nitrate nitrogen to 75% urea nitrogen (T12) had the most substantial impact, leading to significant increases in plant height, stem thickness, and leaf area compared to other treatments. Following closely was the T4 treatment. After 60 days of fertilization, the T12 treatment resulted in remarkable growth improvements, with plant height, stem thickness, and leaf area increasing by 63.70%, 32.59%, and 76.14%, respectively, compared to the control (CK) treatment. Similarly, the T4 treatment exhibited substantial growth enhancements, with increases of 50.61%, 29.69%, and 62.43% in plant height, stem thickness, and leaf area, respectively, compared to the CK treatment.

Compared to treatments utilizing a single form of nitrogen (T1–T3), the combined nitrogen form treatments (T4–T12) significantly promoted root development in tomato plants. Moreover, when comparing the impact of treatments using ammonium nitrogen (AN) and urea nitrogen (UN) to those using nitrate nitrogen (NN) in combination with AN or UN, it was evident that the latter combinations led to increased root length (Figure 2A), total root surface area (Figure 2B), total root volume (Figure 2C), number of root tips (Figure 2D), and root activity (Figure 2E). These findings highlight the root-growth-enhancing properties of ammonium nitrogen. However, it is noteworthy that excessive soil ammonium nitrogen content, exceeding 50%, had an inhibitory effect on root development. As ammonium nitrogen levels increased, root length, surface area, volume, and the number of root tips decreased. Among the treatments, the tomato plants treated with a nitrogen ratio of 25%NN to 75%UN (T12) exhibited the highest total root length (2879.80 cm), root volume (9.32 cm^3^), and number of root tips (4965.67). Meanwhile, those treated with 75%NN and 25%AN (T4) had the greatest root surface area (557.47 cm) and root activity (101.69 mg·g^−1^·h^−1^) (Figure 2). However, no significant differences were observed between the T4 and T12 treatments. Furthermore, the exclusive AN fertilization treatment (T2) led to decreases in root length, surface area, volume, and activity by 9.51%, 19.26%, 48.48%, and 25.26%, respectively, compared to the control (CK). Overall, the analysis suggests that treatments with 75%NN and 25%AN, as well as 25%NN and 75%UN, were the most effective in promoting the growth and development of tomato roots.

### 3.2. Effects of the Different Forms of Nitrogen on Photosynthesis

#### 3.2.1. Gas Exchange and Chlorophyll Fluorescence Parameters 

Gas exchange parameters serve as a more direct indicator of a plant’s photosynthetic capacity. The Pn (Figure 3A), Gs (Figure 3B), and Tr (Figure 3C) were highest with the T4 treatment, whereas the Ci (Figure 3D) was lowest with the T12 treatment, followed by T4 and T11. The Pn and Gs were lowest with the AN and UN (T7–T9) treatments, while Tr was lowest with both the T2 and T9 treatments, and was significantly lower than the results for the other treatments. The Ci was significantly higher with the T2 treatment when compared with all other treatments (Figure 3). Overall, the T4 treatment significantly promoted the gas exchange parameters in the tomato leaves. Chlorophyll fluorescence parameters offer insights into a plant’s photosynthetic mechanisms and physiological condition, revealing the intricate connection between photosynthesis and the environment. The various nitrogen treatments significantly increased the maximum photosynthetic efficiency Fv/Fm (Figure 3E), photochemical quenching coefficient qP (Figure 3G), and actual photosynthetic efficiency Y(II) (Figure 3H) of photosystem II in tomato leaves. Within this context, the T4 treatment showed the most substantial increase in both Fv/Fm and qP, whereas the T12 treatment exhibited the most significant improvement in Y(II). On the other hand, the non-photochemical quenching coefficient (NPQ) of photosystem II decreased significantly in tomato leaves treated with T4 and T12 (Figure 3F). In contrast, the T9 treatment saw a significant increase in NPQ when compared to the CK, with no remarkable differences noted among the other treatments. Notably, the color variation observed in the image matched the corresponding parameter changes (Figure 4). 

#### 3.2.2. Expression of Key Enzyme Genes from the Calvin Cycle

The photosynthetic capacity of plants ultimately depends on the CO_2_ assimilation and regeneration abilities of RuBP. To study the effects of different forms of nitrogen on photosynthesis in tomato leaves, key enzymes such as Rubisco, GAPDH, FBA, FBPase, and TK, which are involved in the Calvin cycle, were analyzed. The activities of Rubisco (Figure 5A), FBA (Figure 5B), GAPDH (Figure 5C), FBPase (Figure 5D), and TK (Figure 5E) in the tomato leaves treated with AN and NN, and with NN and UN, were significantly increased. Specifically, the activities of Rubisco, GAPDH, FBA, FBPase, and TK in the tomato leaves treated with 25%NN:75%UN (T12) and 25%AN:75%NN (T4) were significantly increased, while their activities in the leaves treated with AN and UN decreased as the AN ratio increased.

To further explore how different forms of nitrogen affect CO_2_ assimilation and RuBP regeneration rates, the effects of the different nitrogen forms on key enzyme genes from the Calvin cycle were assessed. The results showed that the expression levels of *SlRbcL* (Figure 6A), *SlRbcS* (Figure 6B), *SlFBA* (Figure 6C), *SlGAPDH* (Figure 6D), *SlFBPase* (Figure 6E), and *SlTK* (Figure 6F) in the tomato leaves were significantly upregulated after treatment with 25%NN:75%UN (T12) and 25%AN:75%NN (T4). Compared with the T4 treatment, the T12 treatment significantly upregulated the expression of *SlRbcL*, *SlRbcS*, *SlGAPDH,* and *SlTK* in tomato leaves. In addition, the combined application of AN and UN significantly decreased the expression of key genes in the Calvin cycle in tomato leaves when compared with the application of NN alone.

### 3.3. Effects of Different Forms of Nitrogen on Sugar Metabolism

#### 3.3.1. Effects on the Photosynthetic Products

The synthesis and accumulation of photosynthetic products in tomato leaves with different nitrogen treatments were investigated (Figure 7). The fructose (Figure 7A), glucose (Figure 7B), and sucrose (Figure 7C) in the tomato leaves treated with AN and NN (T4–T6) or NN and UN (T10–T12) were significantly higher than those treated with single N forms or the AN and UN combination. The starch (Figure 7D) in the tomato leaves treated with NN and AN, or with NN and UN significantly decreased, among which the T12 decreased most significantly, followed by T5. Compared with the other treatments, T12 showed the most obvious increase in fructose, glucose, and sucrose content, followed by T4 and T5. These results show the effects of different forms of nitrogen on the synthesis and accumulation of photosynthetic products in tomato leaves, and that accumulation was significantly promoted by the combined applications of AN and NN or NN and UN.

#### 3.3.2. Sugar-Metabolism-Related Enzymes and Gene Expression

Sugar is a product of photosynthesis and plays a crucial role in overall plant metabolism. In this study, we investigated the activity of enzymes associated with sugar metabolism in tomato leaves subjected to various nitrogen treatments (as depicted in Figure 8). Notably, the concurrent application of NN and UN (T10–T12) or AN and NN (T4–T6) led to a notable increase in the activities of SS (Figure 8A), SPS (Figure 8B), AI (Figure 8C), NI (Figure 8D), α-amylase (Figure 8E), and β-amylase (Figure 8F) within the tomato leaves. Conversely, when AN and UN were combined, there was a reduction in the enzymatic activity related to sugar metabolism in the tomato leaves. Among these enzymes, SS, SPS, NI, α-amylase, and β-amylase exhibited their highest activities with the 25%NN:75%UN treatment (T12), whereas AI activity peaked with the 25%AN:75%NN treatment (T4). Interestingly, tomato leaves treated with a single AN application (T2) showed the lowest enzyme activities associated with sugar metabolism, with no significant differences observed between the other treatments.

The relative expression of genes encoding sugar-metabolism-related enzymes was analyzed in the tomato leaves treated with different forms of nitrogen. The AN and NN (T4–T6) and NN and UN (T10–T12) fertilization treatments significantly enhanced the expression levels of *SlSS* (Figure 9A), *SlSSP* (Figure 9B), *SlAI* (Figure 9C), *SlNI* (Figure 9D), *Slα-amylase* (Figure 9E), and *Slβ-amylase* (Figure 9F). The expression levels of *SlSS*, *SlSSP, SlAI*, *SlNI*, *Slα-amylase,* and *Slβ-amylase,* however, were significantly decreased after the AN and UN (T7–T9) treatments. In addition, the expression of genes encoding sugar-metabolism-related enzymes was greatest with the 75%NN:25%AN (T4) and 75%UN:25%NN (T12) treatments. The results showed that NN combined with an appropriate amount of AN or UN could significantly upregulate the expression of *SlSS*, *SlSSP*, *SlAI*, *SlNI*, *Slα-amylase*, and *Slβ-amylase* and promote the synthesis and accumulation of photosynthetic products in tomato leaves.

### 3.4. Effects of Different Forms of Nitrogen on the Metabolism of Nitrogen in Tomato Leaves

Various nitrogen treatments exerted significant effects on the activities of key enzymes involved in nitrogen metabolism within tomato leaves, as depicted in Figure 10. Notably, the activities of NR (Figure 10A), NiR (Figure 10B), GS (Figure 10C), GOGAT (Figure 10D), and GDH (Figure 10E) reached their peak in tomato leaves when subjected to a combination of 25%NN:75%UN (T12) or 25%AN:75%NN (T4). Comparatively, employing a single nitrogen source yielded lower activities of NR, NiR, GS, GOGAT, and GDH. In particular, the combined application of NN with AN or UN resulted in an enhancement of NR, NiR, GS, GOGAT, and GDH activities. Conversely, when AN and UN were combined, a decrease in NR, NiR, GS, and GOGAT activities was observed. Remarkably, GDH activity exhibited a significant increase when a substantial proportion of AN or a dose of AN alone was administered. These findings indicate that the synergistic combination of NN with AN or UN promotes the GS-GOGAT cycle of nitrate reduction, thereby augmenting nitrogen metabolism within tomato leaves.

To assess the relative expression levels of nitrogen-metabolism-related genes in tomato leaves under different nitrogen sources, we employed qRT-PCR analysis. The results unveil noteworthy insights: *SlNR* and *SlNiR* expression levels experienced significant upregulation when subjected to the 25%NN:75%UN (T12) and 25%AN:75%NN (T4) treatments, as illustrated in Figure 11A,B. However, as the AN application exceeded 50%, *SlNR* expression markedly decreased, and followed a similar trend for *SlNiR*. The expression of *SlGS* displayed a substantial increase upon combining NN with UN treatment, while it exhibited a significant decline when subjected to NN and AN or UN treatments, as indicated in Figure 11C,D. Likewise, the expression of SlGOGAT decreased significantly after the AN and NN or UN treatments, but it increased significantly when NN and UN were combined (Figure 11E). Moreover, NN and UN treatments alone led to increased *SlGOGAT* expression levels. In addition, *SlGDH* expression levels were significantly upregulated with a single AN dose application. However, as the ammonium nitrogen application exceeded 50%, SlGDH expression showed a notable increase. Moreover, when the AN treatment was combined with NN or UN, SlGDH expression was also significantly upregulated (Figure 11F).

To determine the effects of the different nitrogen treatments on the Calvin cycle, glucose metabolism, and nitrogen metabolism, principal component analysis was performed to assess the activities of 16 enzymes involved in metabolism (Figure 12A). The results showed significant differences in the enzyme activities involved in the metabolism of the three nitrogen forms. PC1 and PC2 accounted for 73.4% and 9.4% of the total variance, respectively. The cumulative contribution rate of the first two principal components accounted for 82.80%, indicating that they were sufficient to distinguish the differences in enzyme activities of the tomato leaves treated with different forms of nitrogen. In addition, CK, T1–T3, and T6–T9 and T4, T5, and T10–T12 showed obvious separations based on PC1, while T4, T5, T10, T11, and T12 showed obvious separations based on PC2. This classification result was supported by a cluster analysis. The classification model based on the cluster analysis also divided the 13 treatments into two categories: T4, T5, T12 and CK, T1–T3, T6–T11 (Figure 12B).

A classification model for nitrogen metabolism, glucose metabolism, and Calvin cycle gene expression was developed for tomato leaves receiving different nitrogen treatments based on the principal component analysis (Figure 13A). The separation of the variables and the differences in metabolic gene expression in the tomato leaves of the 13 treatments were highlighted. The two principal components explained 91.6% of the total variance, of which PC1 and PC2 accounted for 80.4% and 11.2%, respectively, indicating that the model can make accurate predictions. The load graph (Figure 13A) shows that the sugar metabolism genes *Slα-amylase*, *Slβ-amylase*, and *SlSS,* and the Calvin cycle genes *SlGAPDH* and *SlRbcS*, had intense first component loads, and *SlGOGAT*, *SlGS1*, and *SlGS2* had powerful second principal component loads. At the same time, CK, T1–T3, T6–T9 and T4, T5, T10–T12 showed an apparent separation based on PC1, and T4, T5, and T10–T12 showed a clear separation based on PC2. Based on the differences in gene expression in the tomato leaves treated with different forms of nitrogen, the treatments were divided into two categories based on the classification model of cluster analysis: T12 and CK and T1–T11 (Figure 13B).

## 4. Discussion

### 4.1. Effects of Different Forms of Nitrogen on Tomato Growth and Root Morphological Parameters

Plant roots possess a multitude of basic self-adaptive capabilities, encompassing water and nutrient absorption, soil anchorage, and the establishment of symbiotic relationships with root-associated biological communities. Consequently, the diverse forms of nitrogen can exert significant impacts on plant root structures [29,30]. In this study, we observed that in comparison to the sole application of a single nitrogen source, utilizing a blend of 25%NN:75%UN (T12) or 75%NN:25%AN (T4) significantly enhanced the primary root length, root surface area, root volume, and the number of root tips in tomato plants, thereby promoting the development of an extensive root system. Conversely, when AN was combined with UN, it hindered the growth of tomato root structures, with a more pronounced inhibition observed when AN application exceeded 50% (Figure 2). Numerous studies have demonstrated that an increasing concentration of NH_4_^+^ in the soil results in a reduced root–shoot ratio, shorter roots, fewer lateral roots, and a deeper coloration, ultimately influencing plant growth and development [31,32]. It was also found that an increase in the AN ratio affected plant growth, as the presence of the NH_4_^+^ greatly delayed the absorption of NO_3_^−^ [33,34,35,36]. Yan et al. [37] elucidated that a higher proportion of ammonium fertilizer could induce rhizosphere acidification, leading to an elevation in extracellular H^+^ concentration and the inhibition of proton pumps in cell membranes. Reduced proton pump activity can lead to cell membrane depolarization, affecting the root system’s ability to absorb other ions and ultimately impeding plant growth. Conversely, increasing the proportion of nitrate nitrogen in the nutrient solution can mitigate or prevent ammonium toxicity. This adjustment can ameliorate rhizosphere acidification resulting from excessive ammonium nitrogen uptake by plants [38]. Kirkby et al. [39] discovered that when AN and UN were mixed, tomato plants exhibited increased lateral root length and lateral root numbers. In our study, 25%AN:75%NN treatment also significantly promoted root development in tomato plants. However, the 25%NN:75%UN treatment was more effective than the 25%AN:75%NN treatment.

In this study, the results show that the three treatments that use NN combined with UN result in significant increases, in line with the UN ratio, in plant height, stem thickness, and leaf area for the tomatoes (Figure 1). It indicated that the high proportion of UN and NN combined application could promote the absorption and utilization of nitrogen in tomato plants, and then improve the growth of tomato plants. The promotion effect of a high proportion of NN combined with UN on tomato growth was very weak; the observed increase in these parameters can be attributed to the enhanced absorption efficiency of urea in the presence of NO_3_^−^. However, the presence of urea does not appear to affect the absorption of NO_3_^−^ [40,41,42]. It has been found that NO_3_^−^-N is more conducive to the growth and development of tomato than NH_4_^+^-N [11]. Compared to the AN (T2) treatment, the NN treatment significantly increased the plant height, stem thickness, leaf area, and root morphological parameters of the tomato plants, and the growth performance of the 75%NN:25%AN (T4) plants was the best. However, when the AN application exceeded 50%, the plant height, stem thickness, leaf area, and root morphological parameters of the tomato plants were significantly reduced, and the application of AN alone resulted in inhibition (Figure 1). The results showed that a suitable AN ratio was beneficial for the growth and development of tomato plants and promoted the absorption and assimilation of nitrogen by crops, whereas a high proportion of AN inhibited the growth of tomato plants. Previous studies have found that [31,32,43], when compared with nitrate nitrogen, ammonium nitrogen results in a smaller leaf area and more significant damage to tomato plants than sugarcane and rice. Zou et al. [44] studied tomato roots under different ammonium supply levels and found that when the NH_4_^+^ supply exceeded 0.5 mmol·L^−1^, the main root length, lateral root number, and plant height decreased with the increase in NH_4_^+^, and 10 mmol·L^−1^ NH_4_^+^ almost completely inhibited lateral root formation, which was consistent with our results. Therefore, the appropriate amount of NH_4_^+^ combined with NO_3_^−^ can increase the absorption and utilization of nitrogen in tomato plants, reduce the toxicity of NH_4_^+^ to tomato, and promote the growth of tomato roots and plants.

### 4.2. Different Forms of Nitrogen Affected the Chlorophyll Fluorescence Parameters and Photosynthesis in the Tomato Leaves

This study showed that, under the same nitrogen application rate, the combination of 25%AN:75%NN (T4) and the combination of 25%NN:75%UN (T12) could significantly increase the photosynthetic rate of the tomato leaves. The net photosynthetic rate, however, was significantly reduced when AN and UN (T7–T9) were applied together or the proportion of AN was more than 50% (Figure 3). This may be due to the combined application of 25%AN and 75%NN or 25%NN and 75%UN having significantly increased the flow of photosynthetic carbon to amino acids, which in turn led to an increase in photosynthetic enzyme content, while a high proportion of ammonium nitrogen inhibited the synthesis of photosynthetic products, resulting in a decrease in photosynthetic carbon content, which is consistent with the results of Golvano et al. [45]. Glaussen et al. [46] and Puritch et al. [47] also found that the accumulation of ammonium nitrogen in plant leaves may lead to the uncoupling of electron transport in chloroplasts to form phosphorylation, which eventually leads to a decrease in photosynthetic rate. Others believe that inhibition of the photosynthetic rate is caused by the toxic effects of ammonium nitrogen [48,49]. Yin et al. [50] found that the sucrose metabolism level of the 75%NO_3_^−^-N:25%NH_4_^+^-N treatment was the highest, and the net photosynthetic rate of the leaves increased significantly in Cabernet Sauvignon (*Vitis vinifera* cv.). Tabatabaei et al. [51] reported that the growth rate of plants was maximized when sucrose was used instead of amino acids as a nutrient supply. This suggests that plants require high net photosynthetic rates to produce photosynthetic products for growth and metabolism, and subsequently to achieve optimal growth. To achieve a higher net photosynthetic rate, it is necessary to maintain a higher nitrogen level and a suitable nitrogen form ratio.

Chlorophyll fluorescence imaging is a valuable tool for measuring photosynthesis in plants. It reflects changes in the thylakoid membrane structure and function, photoinhibition, and O_2_ release by interacting with PSII components [52,53]. In addition, it can accurately determine the activity and dynamic changes in PSII, reflect all aspects of photosynthesis, and detect the effects of stress on photosynthesis [54]. Fv/Fm represents the original light–energy conversion efficiency of PSII. In one study, the decrease in Fv/Fm indicated that the PSII reaction center was damaged and that the plant was inhibited by light [55]. In another study, the Fv/Fm changed little under normal conditions and decreased significantly when plants were subjected to stress [56]. Our study reveals that the treatments involving 25%NN:75%UN (T12) and 25%AN:75%NN (T4) consistently maintained a relatively high Fv/Fm value, as depicted in Figure 3E. However, when compared to treatments using a singular nitrogen source, the combination of a substantial amount of ammonium nitrogen with a complete lack of nitrogen treatment significantly reduced the Fv/Fm ratio in tomato plants. This observation suggests that an appropriate concentration of ammonium nitrogen can effectively sustain the potential photosynthetic activity of Photosystem II (PSII) in tomato leaves. In contrast, a high proportion of ammonium nitrogen can potentially damage the PSII reaction center, leading to photoinhibition in plants. These findings align with those reported by Krause et al., reinforcing the consistency of our results with prior research [57]. Y(II) represents the quantum yield of photosynthetic electron transport in plants and reflects the rate of photosynthetic electron transport in the leaves [58]. In this study, the 25%NN:75%UN (T12) treatment increased the Y(II) of the tomato leaves, whereas no nitrogen application (CK) and a single AN treatment significantly reduced Y(II) (Figure 3H). There were no significant differences between the other treatments, which is consistent with the changes in plant photosynthesis introduced previously. The decrease in Y(II) can be attributed to the degradation of the lamellar structure of the base grains. This degradation reduces the light energy harvesting area and decreases the total electron yield [59]. Rehab et al. [60] found that salt stress leads to the deformation of the thylakoid membrane and disintegration of the grana structure, thus reducing the photosynthetic performance of plants. Singh et al. [61] found that nitrate nitrogen sources can protect photosynthetic pigments from ammonium nitrogen stress in tomato leaves, which competitively reduces the absorption of ammonium nitrogen by tomato plants. Moreover, the mixed application of nitrate nitrogen with urea or ammonium nitrogen had the best effect, and the high concentration of nitrogen source (15 mmol·L^−1^) was more conducive to the growth and development of tomato seedlings. qP represents the fraction of light energy absorbed by PSII antenna pigments for photochemical electron transfer. This indicates the extent to which the PSII reaction center was open [62]. In this study, we observed no significant difference in the proportion of PSII allocated to photochemical electron transfer in tomato leaves when combining AN and NN (T4–T6) or when combining AN and UN (T7–T9). However, a notable reduction in the proportion of PSII allocated to photochemical electron transfer was evident after the application of a single nitrogen source. Conversely, the combination of NN and UN (T10–T12) led to a significant increase in the proportion of electrons engaged in photochemical reactions, as illustrated in Figure 3G. It is worth noting that prior research has elucidated that NPQ plays a role in maintaining the high oxidation state of PSII primary electron acceptors, consequently lowering the risk of photodamage during photosynthesis [63,64]. In this study, NPQ changed significantly, and the energy dissipation of 25%AN:75%NN (T4), NN, and UN (T10–T12) decreased significantly (Figure 3F). It was thus concluded that tomato leaves treated with 25%AN:75%NN (T4) or NN: UN (T10–T12) can obtain more light energy and change the direction of light energy absorbed in PSII, which was consistent with the results of Osório et al. [65]. Nasraoui-Hajaji et al. [66] observed that tomato plants receiving ammonium nitrogen or amide nitrogen exhibited lower rates of CO_2_ assimilation, stomatal conductance, and transpiration compared to those supplied with exclusively NO_3_^−^-N as the nitrogen source. Furthermore, the inclusion of nitrate in the nitrogen supply mitigated the reduction in these parameters. These findings suggest that a combination of low ammonium and high NO_3_^−^-N supply is conducive to promoting the growth and development of tomato plants [46]. In our study, it was found that the application of the 75%UN:25%NN (T12) ratio significantly increased the gas exchange parameters, chlorophyll fluorescence parameters, and net photosynthetic rate of tomato plants, and the effect was better than that of the 75%NN:25%AN (T4) ratio. Therefore, the effects of different nitrogen forms on photosynthesis of tomato were different. The application of 75%UN:25%NN (T12) could obviously enhance the maximum photochemical quantum yield (Fv/Fm), the actual photochemical quantum yield Y(II), and the photochemical quenching coefficient (qP) of plant leaves. It can also reduce the NPQ, thereby improving the efficiency of plant absorption, utilization, and distribution of light energy.

### 4.3. Different Forms of Nitrogen Affect the Activity of Key Enzymes in the Calvin Cycle and the Relative Expression of Genes in Tomato Leaves

The photosynthetic rate was influenced by the activity of Rubisco in the carboxylation reaction stage and the regeneration capacity of RuBP. The regenerative ability of RuBP relies on electron transport chain reactions to produce ATP and NADPH, which provide energy for the regeneration phase of the Calvin cycle [67]. This study demonstrated that treatments with a ratio of 25%NN:75%UN (T12) and 25%AN:75%NN (T4) significantly increased the activities of Rubisco, GAPDH, FBA, FBPase, and TK in tomato leaves (Figure 5). However, the activities of Rubisco, GAPDH, FBA, FBPase, and TK decreased as the AN ratio increased in tomato leaves treated with AN and UN, consistent with the findings of previous research conducted by Raab et al. [68]. According to our study, the relative expression levels of genes associated with the Calvin cycle, including *SlRbcL*, *SlRbcS*, *SlFBA*, *SlGAPDH*, *SlFBPase*, and *SlTK*, in tomato leaves treated with a combination of 25%NN:75%UN (T12) or 25%AN:75%NN (T4), exhibited a noteworthy increase. Furthermore, there was a significant elevation in the activity of enzymes involved in the Calvin cycle. However, when a substantial proportion of AN was applied, it led to the inhibition of gene expression and a reduction in the activity of enzymes associated with the Calvin cycle (Figure 6). This phenomenon may be attributed to ammonium toxicity limiting the activity of Rubisco, which, in turn, results in decreased photosynthetic capacity and subsequently impacts the light utilization and overall performance of tomato plants [69,70]. In this study, single NN, AN, and UN treatments reduced the expression of *SlRbcL* and *SlRbcS* genes. The addition of UN increased the expression of *SlRbcL* and *SlRbcS* genes. Therefore, we believe that UN can regulate the effect of a single nitrogen form on the activity of photosynthesis-related enzymes, thereby enhancing the photosynthetic capacity of tomato plants [71]. This finding is consistent with previous studies on tobacco [72] and cucumbers [73]. Researchers have discovered that the upregulation of genes involved in the Calvin cycle can increase the net photosynthetic rate of plants, thereby promoting their growth. Conversely, the downregulation of these genes can impede plant growth. In summary, different forms of nitrogen affect the Calvin cycle gene expression levels and enzyme activities. However, we found that urea could not only reduce the effect of ammonium toxicity on tomato plants, but also significantly increase the enzyme activity and relative gene expression of the Calvin cycle involved in the photosynthesis of tomato plants through the use of NN combined with UN. In particular, the expression of Rubisco size subunit was elevated, which promoted the photosynthetic carbon cycle and organic matter synthesis. This provided a material basis for the plant to develop a larger leaf area and biomass, which ultimately promoted the growth of tomato.

### 4.4. Effects of Different Forms of Nitrogen on Photosynthetic Products in Tomato Leaves

Nitrogen is essential for plant growth and development. Its application level and nitrogen form can significantly affect carbon metabolism in plants [74]. Studies have shown that nitrate nitrogen promotes soluble sugar accumulation in plants, while ammonium nitrogen is more conducive to starch accumulation in the leaves [75]. Carbohydrates, being the primary or advanced products of photosynthesis, serve as useful indicators for measuring the net photosynthetic rate in plants. Our study demonstrated that treatments involving 25%NN:75%UN (T12) and 25%AN:75%NN (T4) significantly increased the content of fructose (Figure 7A), glucose (Figure 7B), and sucrose (Figure 7C) within tomato leaves. Conversely, the combined application of AN and UN (T7–T10) led to starch accumulation in tomato leaves, accompanied by reductions in fructose, glucose, and sucrose levels (Figure 7D). The effects of different nitrogen forms on total soluble sugar and starch contents in tomato leaves were consistent with the changes in the net photosynthetic rate. These findings are in line with research conducted by Zhang et al. [76] on chrysanthemum, where they observed that the 25%NN:75%AN treatment resulted in the highest total soluble sugar content. This suggests that various plants may exhibit distinct responses to different nitrogen forms in terms of total soluble sugar and starch content, potentially due to their sensitivity to nitrogen forms. However, our findings were different from previous studies. We found that 25%NN:75%UN treatment had a better effect on the synthesis of tomato photosynthetic products than 25%AN:75%NN treatment. This may be related to the sensitivity of different plants to nitrogen forms.

Furthermore, our investigation delved into the activities of enzymes and expression levels of genes associated with sugar metabolism. The results revealed that the combined application of NN and UN (T10–T12) or AN and NN (T4–T6) increased the activities of SS (Figure 8A), SPS (Figure 8B), AI (Figure 8C), NI (Figure 8D), α-amylase (Figure 8E), and β-amylase (Figure 8F), all of which play roles in sugar metabolism in tomato leaves. In contrast, the combination of AN and UN (T7–T10) treatments decreased the activity of enzymes involved in sugar metabolism in tomato leaves. Among the various treatments, tomato leaves treated with 25%NN:75%UN (T12) exhibited the highest activities of SS, SPS, NI, α-amylase, and β-amylase. Conversely, tomato leaves treated with 25%AN:75%NN (T4) displayed the highest AI activity among the enzymes related to sugar metabolism. Notably, the activity of sugar-metabolism-related enzymes was lowest in tomato leaves treated with a single AN source (T2). Additionally, the relative expression levels of genes encoding sugar-metabolism-related enzymes followed a similar pattern as enzyme activity, corroborating the findings of Yin et al. [54]. The size of leaf area can directly reflect the nitrogen supply and can be used to evaluate any species [77]. The leaf area of tomato plants treated with 25%NN:75%UN was larger than that of those treated with other nitrogen forms. The larger the leaf area, the more light-energy absorbed by the leaves. This effect was consistent with the increase in the accumulation of photosynthetic product content after mixed application of 25%NN:75%UN treatment. The mixed nitrogen fertilization treatments, particularly the 25%NN:75%UN (T12) treatment, enhanced the activity of sugar metabolism enzymes and gene expression in tomato leaves (Figure 9). In turn, this promoted starch metabolism and the accumulation of total soluble sugars. The photosynthetic efficiency of tomatoes can be optimized through the regulation of nitrogen fertilizer production.

### 4.5. Effects of Different Forms of Nitrogen on the Metabolism of Nitrogen in Tomato Leaves

Studies on *ryegrass* [78], *Salvinia natans* [79], and *Canna indica* [80] have confirmed that the activities of NR and NiR in higher plants usually depend on nitrate availability. This experiment showed that the activities of NR and NiR were greatest with the 25%NN:75%UN (T12) treatment, followed by the 75%NN:25%AN (T4) treatment. The NN combined with an appropriate amount of AN or UN can thus be used to improve nitrate reductions in tomato leaves (Figure 10A,B). GS is a key enzyme involved in ammonium assimilation. It catalyzes the synthesis of glutamine from ammonium and glutamic acid. In addition to preventing excess ammonium ions from poisoning the body, glutamine is also the primary storage and transportation form of ammonium. An increase in glutamine levels catalyzes glutamic acid synthase activity and produces glutamic acid [81,82]. This study found that when NN was combined with UN (T10–T12), the GS and GOGAT activity significantly increased, and was the highest after the 25%NN:75%UN (T12) treatment. However, NN combined with AN (T4–T6), AN combined with UN (T10–T12), and the single nitrogen fertilizer (T1–T3) all decreased the GS and GOGAT activities (Figure 10A,B). The 25%NN:75%UN (T12) treatment consequently increased the activity of GS, promoted the assimilation of NH_4_^+^, and eliminated the toxic effects of NH_4_^+^ on plants. Ma et al. [83] found that when compared with 50%CO(NH_2_)_2_: 50%NH_4_^+^-N, a 50%CO(NH_2_)_2_: 50%NO_3_^−^-N treatment significantly enhanced GOGAT and GS enzyme activities and promoted NH_4_^+^ assimilation. However, we found that the promotion effect of 25%NN:75%UN treatment on the NH_4_^+^ assimilation of tomato plants was better than that of 50%NN:50%UN treatment, which may have been caused by the different varieties used. Most studies of nitrogen metabolism have focused on NR, GOGAT, and GS. However, few studies have investigated the effects of nitrogen on glutamate dehydrogenase. GDH catalyzes a reversible reaction and is a key enzyme linking the GS−GOGAT cycle to the tricarboxylic acid cycle. The results of this experiment showed that GDH activity was significantly upregulated after the application of a single AN treatment. When the proportion of AN application decreased, the GDH activity showed a downward trend, indicating that AN plays an essential role in NH_4_^+^ assimilation (Figure 10E). Because of the special role of GDH in nitrogen metabolism, this process is closely related to carbon metabolism [84].

After plants absorb nitrate nitrogen, it must be reduced to ammonium nitrogen before it can be used by the plants. This process requires the participation of NR and NiR. NR is a key rate-limiting enzyme in this process and is substrate-induced. Its activity often increases with an increase in the substrate nitrate [59]. The relative expression of six key genes involved in nitrogen metabolism was also analyzed. This study found that the expression levels of the *SlNR* (Figure 11A) and *SlNiR* (Figure 11B) genes involved in nitrate reduction were upregulated after treatment with 75%NN:25%AN (T4) or 25%NN:75%UN (T12). This results in an increase in NR activity and promotes nitrate assimilation in tomato leaves. These findings align with the research results on corn conducted by Alexander [85]. When AN and UN (T7–T9) were applied together or after a single AN (T2) treatment, the expression levels of the *SlNR* (Figure 11A) and *SlNiR* (Figure 11B) genes were significantly downregulated. Numerous studies have demonstrated that the NR activity decreases with increases in ammonium nitrogen in the nutrient solution [86]. The three forms of nitrogen share a common metabolic pathway known as the glutamate pathway. This pathway converts NH_4_^+^ into amino acids. In our study, when the amount of nitrogen was the same, mixed-form nitrogen exhibited higher GOGAT and GS activities than single-form nitrogen. Additionally, the *SlGOGAT* (Figure 11E), *SlGS1* (Figure 11C), and *SlGS2* (Figure 11D) genes were all significantly upregulated with the mixed-form nitrogen treatment when compared to the single-form nitrogen treatment. Notably, the treatments with 75%NN:25%AN (T4) or 75%UN:25%NN (T12) showed the most significant increases in the expression of genes involved in the glutamate cycle in tomato leaves. These results suggested that mixed nitrogen promotes the glutamate cycle. In nitrogen tests using cucumber [87] and cotton seedlings [88], similar conclusions were also drawn from a study investigating the effects of NN when combined with AN treatment on GS/GOGAT. However, interestingly, in our study, 75%UN:25%NN treatment was more effective than 75%NN:25%AN treatment in improving nitrogen-metabolism-related enzyme activity and gene expression. Ma et al. [83] observed that the expression of *SlGOGAT*, *SlGS1*, and *SlGS2* was downregulated when cells were treated with ammonium nitrogen instead of nitrate nitrogen. Conversely, the application of an equivalent amount of CO(NH_2_)_2_ to nitrate nitrogen promoted nitrogen metabolism. However, when equal amounts of ammonium nitrogen were applied to the nitrate nitrogen, nitrogen metabolism was inhibited. This study showed that in contrast to GS/GOGAT, the expression of the *SlGDH* gene was higher when a single form of nitrogen was used than when mixed nitrogens were used. Similarly, GDH activity was higher with the AN treatment alone, without any significant differences from the other nitrogen treatments. These findings indicate that *SlGDH* (Figure 11E) plays a crucial role in nitrogen metabolism when the GS-GOGAT cycle is inhibited. Numerous studies have consistently demonstrated that plants experiencing growth stress, mainly due to ammonium poisoning, exhibit a significant enhancement in the role of GDH in the nitrogen metabolism pathway. Horchani F. et al. [89] found that with the increase in external NH_4_^+^ concentration, the GDH activity showed an upward trend and *SlGDH* expression was also upregulated. In our study, the enzyme activity and gene expression of GDH were the highest after single AN treatment, and the enzyme activity of GDH decreased with the decrease in the AN application ratio. However, under the condition of 75%UN:25%NN treatment, the GS activity of tomato leaves increased significantly, and the gene expression levels also increased. This indicated that the metabolism of NH_4_^+^ in tomato plants was mainly dependent on the GS-GOGAT pathway at normal levels. GDH may play an important role in metabolizing excessive ammonium and reducing ammonium toxicity in plants [89,90].

Nitrogen fertilizer plays a pivotal role in influencing the growth and development of tomato plants. As the global tomato cultivation area continues to expand, farmers often resort to excessive nitrogen fertilizer applications in a bid to boost yields. However, the indiscriminate choice of nitrogen fertilizers not only hinders the effective absorption and utilization of nitrogen by tomato plants but also exposes them to potential toxicity, with adverse environmental consequences. In summary, nitrogen blends of 75%UN:25%NN or 75%NN:25%AN were demonstrated to significantly enhance various growth parameters in tomato plants. These effects encompassed increased plant height, stem diameter, leaf area, and root morphological attributes. Furthermore, these nitrogen ratios were also found to enhance gas exchange parameters, chlorophyll fluorescence parameters, and the accumulation of photosynthetic products in tomato leaves. These result in an overall improvement in leaf photosynthetic capacity. Conversely, the combination of AN with UN was shown to inhibit the growth of tomato plants. Additionally, the treatment involving 75%UN:25%NN significantly elevated the activities of enzymes associated with nitrogen metabolism, the Calvin cycle, and sugar metabolism, along with their corresponding gene expression levels (Figure 14). This suggests that 75%UN:25%NN can upregulate the expression of genes related to nitrogen metabolism, the Calvin cycle, and sugar metabolism, consequently increasing enzyme activities. This, in turn, promotes a higher net photosynthetic rate and greater accumulation of photosynthetic products in tomato plants. In contrast, a high proportion of AN was observed to restrain root growth, subsequently impeding the overall growth and development of tomato plants. From a grower’s perspective, it is advisable to consider the utilization of NN and UN in the fertilizer regimen to reduce the reliance on AN. This study provides a valuable practical foundation for tomato cultivation and sustainable agricultural fertilization management in the future.

## Figures and Tables

**Figure 1 plants-12-04175-f001:**
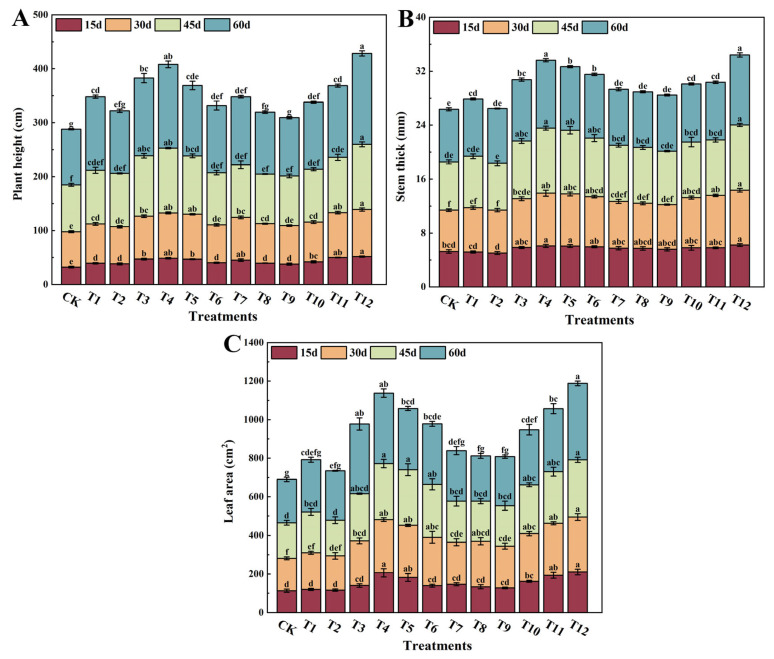
Effects of different nitrogen forms on plant height (**A**), stem thickness (**B**), and leaf area (**C**) of tomato plants. The short vertical line of the bar chart indicates the average value ± standard error (*n* = 5), and the different letters indicate a significant difference at the *p* < 0.05.

**Figure 2 plants-12-04175-f002:**
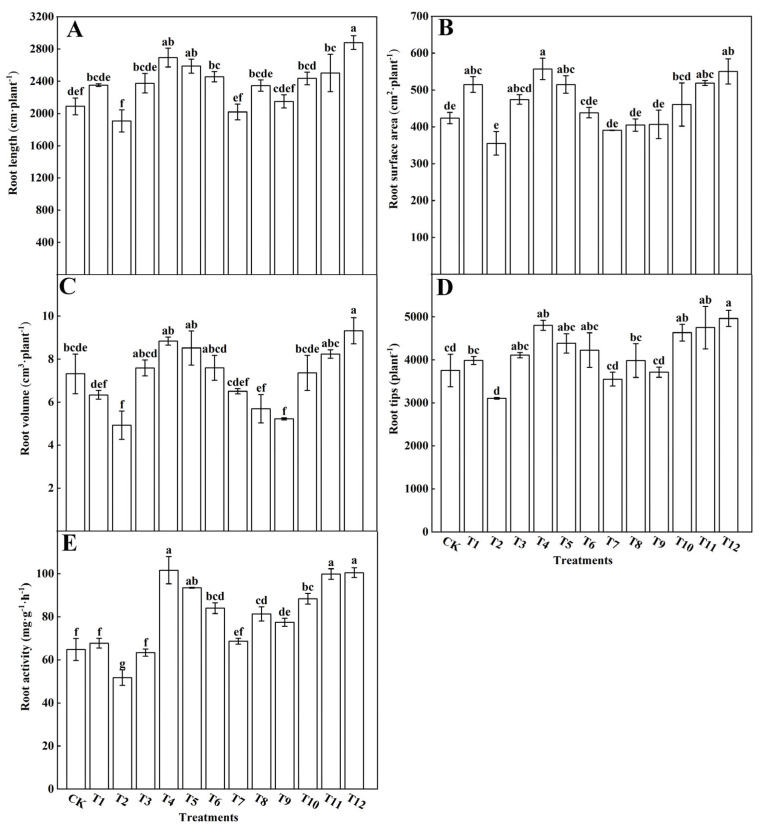
Effects of different forms of nitrogen on tomato root parameters. Root length (**A**), root surface area (**B**), root volume (**C**), root tips number (**D**), root activity (**E**). The short vertical line of the bar chart indicates the average value ± standard error (*n* = 3), and different letters in the same column indicate significant differences at the *p* < 0.05.

**Figure 3 plants-12-04175-f003:**
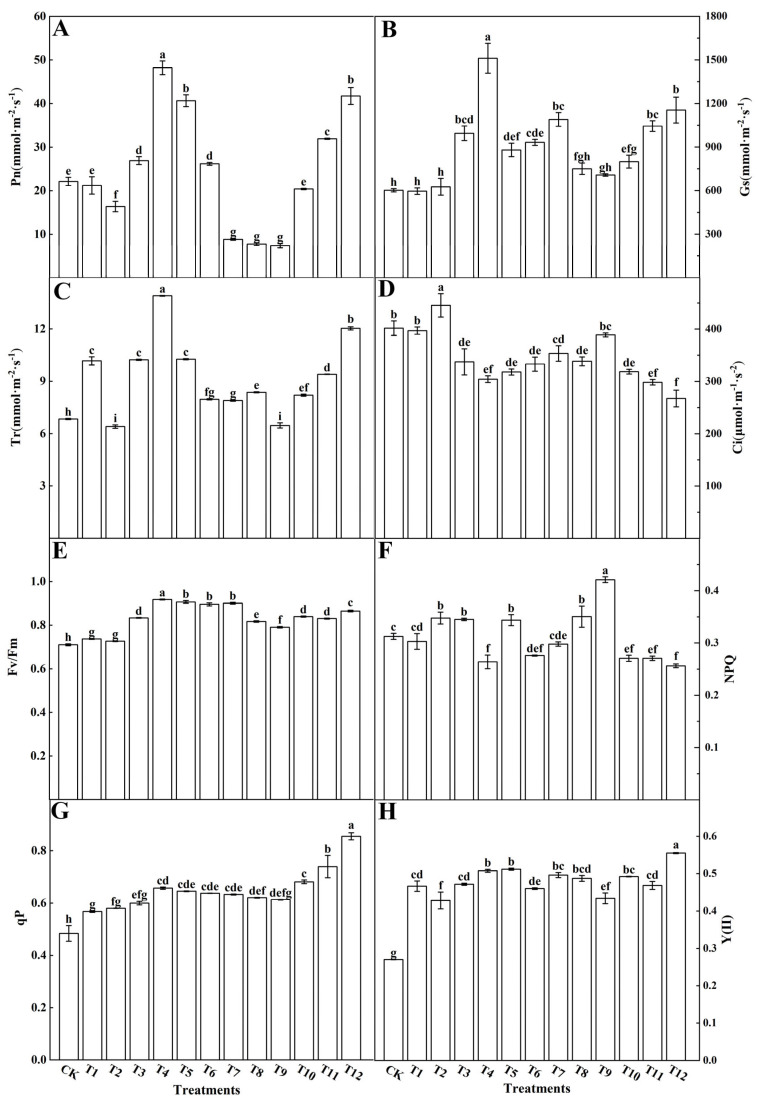
Effects of different nitrogen forms on the net photosynthetic rate (Pn (**A**)), stomatal conductance (Gs (**B**)), transpiration rate (Tr (**C**)), intercellular CO_2_ concentration (Ci (**D**)), maximum quantum yield (Fv/Fm (**E**)), non-photochemical quenching coefficient (NPQ (**F**)), photochemical quenching coefficient (qP (**G**)), and actual photochemical efficiency (Y(II) (**H**)) of tomato leaves. The short vertical line of the bar chart indicates the average value ± standard error (*n* = 3), and the different letters indicate a significant difference at the *p* < 0.05.

**Figure 4 plants-12-04175-f004:**
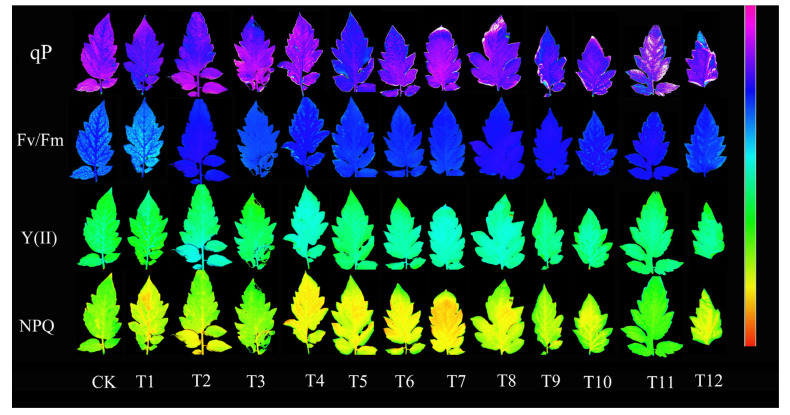
Effects of the different forms of nitrogen on chlorophyll fluorescence imaging of tomato leaves: qP, photochemical quenching coefficient; Fv/Fm, PSII maximum quantum yield; Y(II), the actual photochemical efficiency of PSII; NPQ, non-photochemical quenching coefficient. Each image in the same column represents the same leaf. The color scale at the bottom represents the value from 0 (black) to 1 (purple).

**Figure 5 plants-12-04175-f005:**
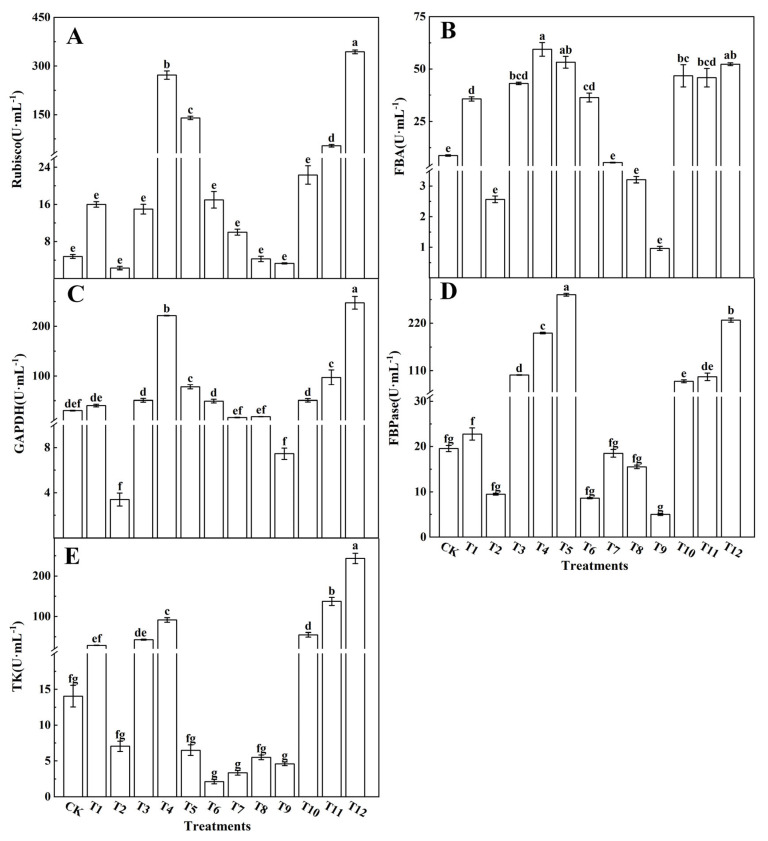
Effects of different nitrogen forms on the activities of ribulose-1,5-bisphosphate carboxylase/oxygenase (**A**), fructose-1,6-bisphosphate aldolase (**B**), glyceraldehyde-3-phosphate dehydrogenase (**C**), fructose-1,6-bisphosphate esterase (**D**), and transketolase (**E**) in tomato leaves. The short vertical line of the bar chart indicates the average value ± standard error (*n* = 3), and the different letters indicate a significant difference at the *p* < 0.05.

**Figure 6 plants-12-04175-f006:**
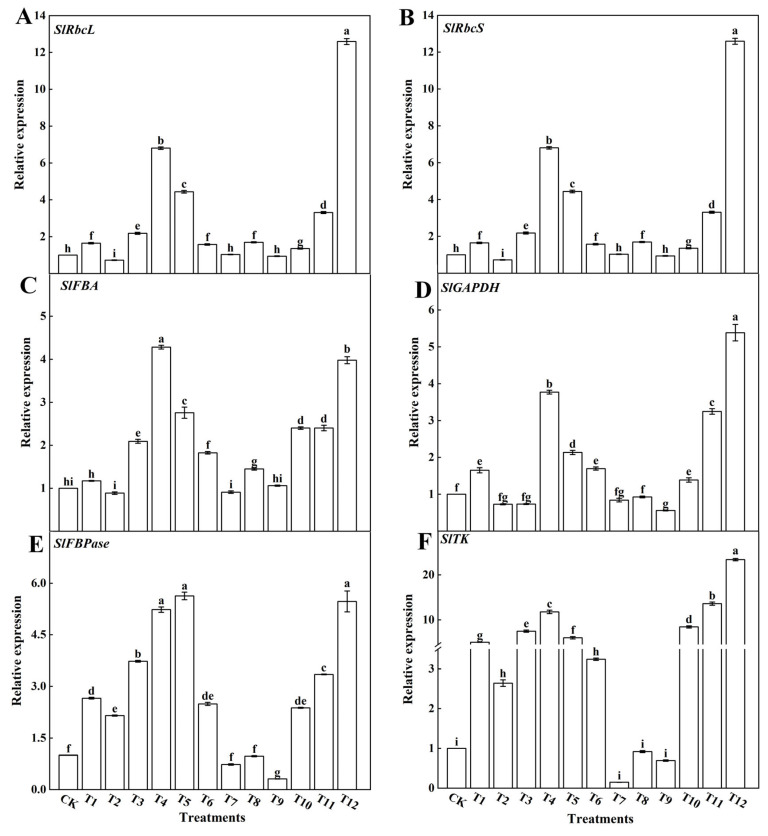
Effects of different nitrogen forms on the key enzyme genes from the Calvin cycle in tomato leaves: Rubisco large subunit (**A**), Rubisco small subunit (**B**), FBA (**C**), GAPDH (**D**), FBPase (**E**), and TK (**F**). The short vertical line of the bar chart indicates the average value ± standard error (*n* = 3), and the different letters indicate a significant difference at the *p* < 0.05.

**Figure 7 plants-12-04175-f007:**
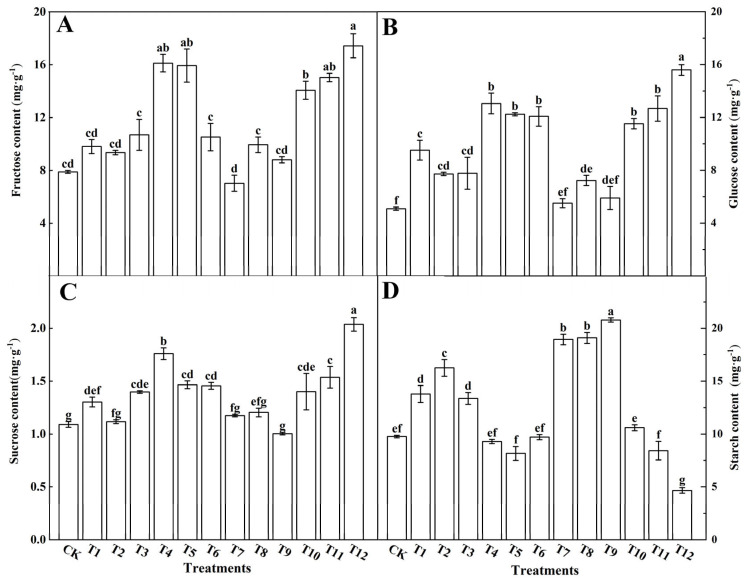
Effects of different nitrogen forms on the fructose contents (**A**), glucose (**B**), sucrose (**C**), and starch (**D**) in tomato leaves. The short vertical line of the bar chart indicates the average value ± standard error (*n* = 3), and the different letters indicate a significant difference at the *p* < 0.05.

**Figure 8 plants-12-04175-f008:**
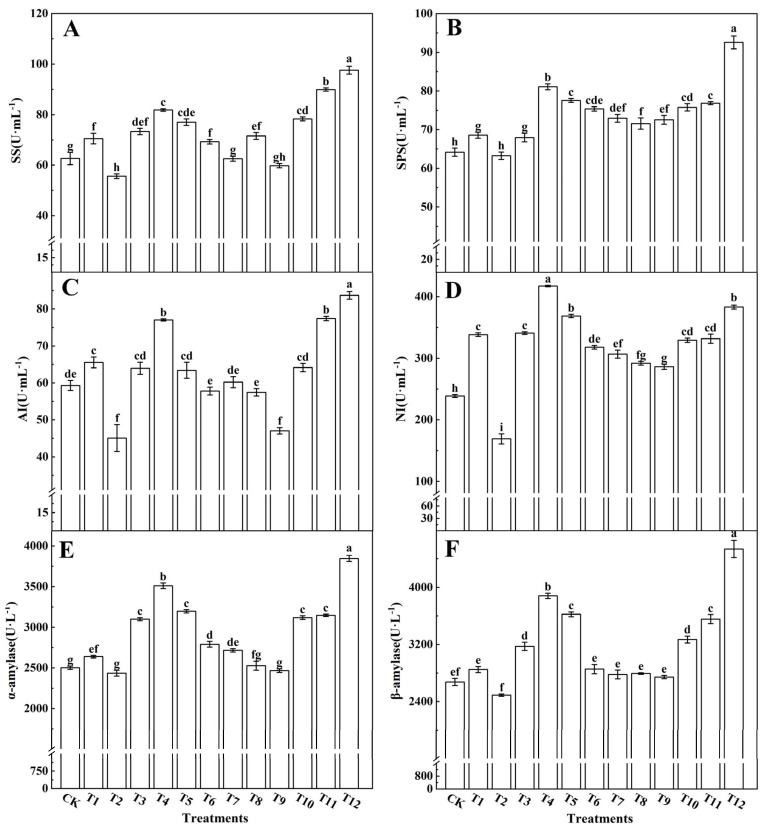
Effects of different nitrogen forms on the sucrose synthase activities (**A**), sucrose phosphate synthase (**B**), acid invertase (**C**), neutral invertase (**D**), α-amylase (**E**), and β-amylase (**F**) in tomato leaves. The short vertical line of the bar chart indicates the average value ± standard error (*n* = 3), and the different letters indicate a significant difference at the *p* < 0.05.

**Figure 9 plants-12-04175-f009:**
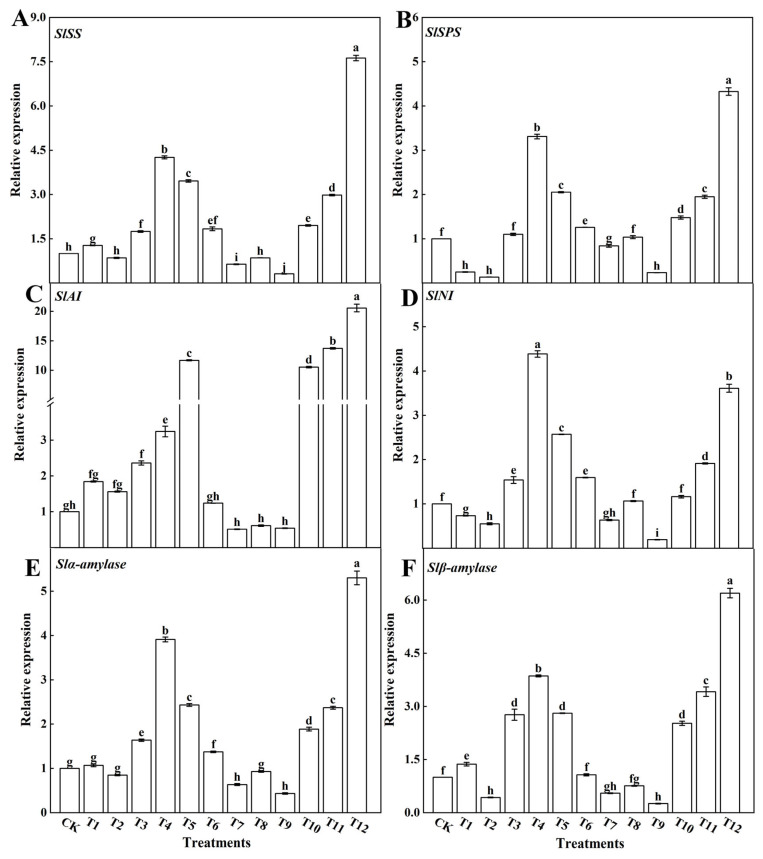
Effects of different nitrogen forms on sugar-metabolism-related enzyme genes in tomato leaves: sucrose synthase (**A**), sucrose phosphate synthase (**B**), acid invertase (**C**), neutral invertase (**D**), α-amylase (**E**), and β-amylase (**F**). The short vertical line of the bar chart indicates the average value ± standard error (*n* = 3), and the different letters indicate a significant difference at the *p* < 0.05.

**Figure 10 plants-12-04175-f010:**
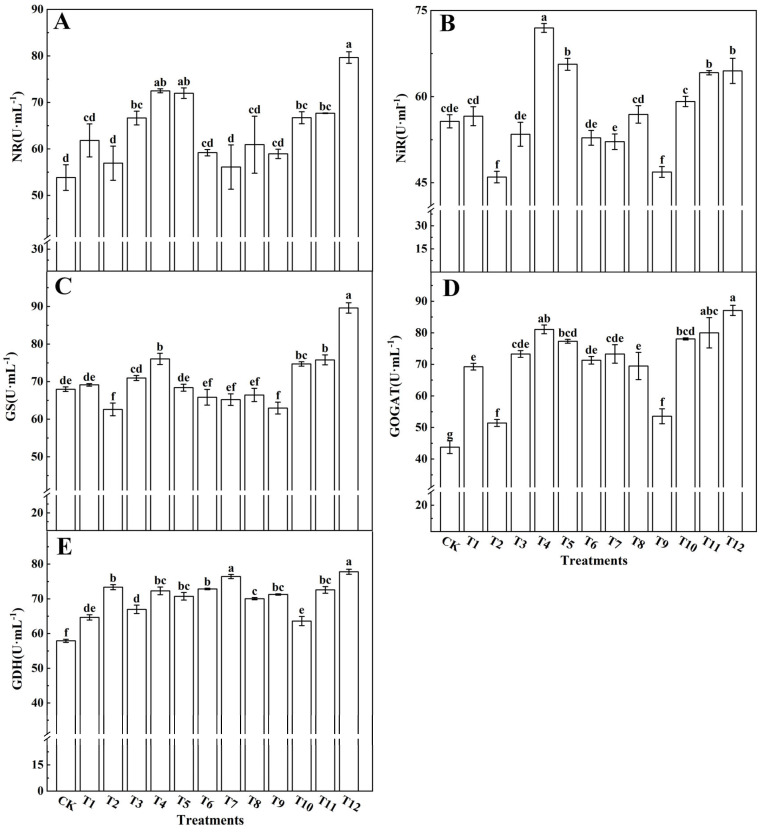
Effects of different nitrogen forms on the activities of nitrate reductase (**A**), nitrite reductase (**B**), glutamine synthetase (**C**), glutamate synthase (**D**), and glutamate dehydrogenase (**E**) in tomato leaves. The short vertical line of the bar chart indicates the mean value ± standard error (*n* = 3), and the different letters indicate a significant difference at the *p* < 0.05.

**Figure 11 plants-12-04175-f011:**
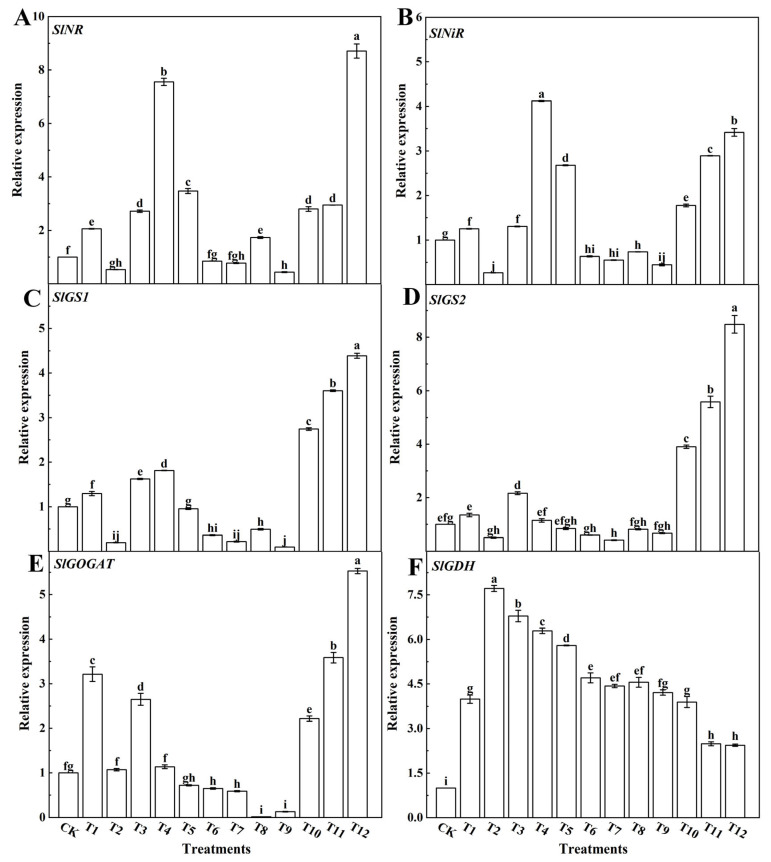
Effects of different nitrogen forms on the expression of *SlNR* (**A**), *SlNiR* (**B**), *SlGS1* (**C**), *SlGS2* (**D**), *SlGOGAT* (**E**), and *SlGDH* (**F**) genes in tomato leaves. The short vertical line of the bar chart indicates the average value ± standard error (*n* = 3), and the different letters indicate a significant difference at the *p* < 0.05.

**Figure 12 plants-12-04175-f012:**
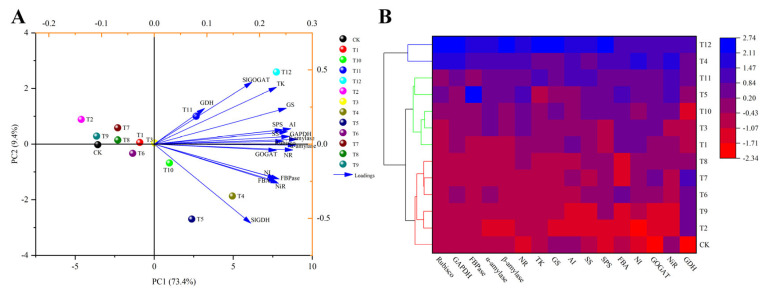
Principal component analysis (**A**) and cluster analysis (**B**) of Calvin cycle, sugar-metabolism-related, and nitrogen-metabolism-related enzyme activities in tomato leaves under different forms of nitrogen treatment.

**Figure 13 plants-12-04175-f013:**
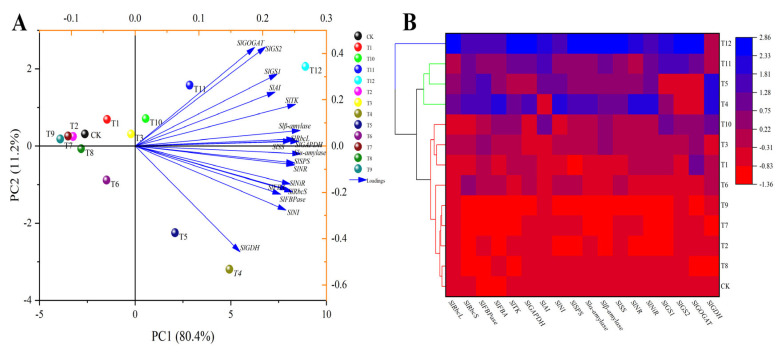
Principal component analysis (**A**) and cluster analysis (**B**) of Calvin cycle, sugar metabolism, and nitrogen metabolism gene expression in tomato leaves under different nitrogen treatments.

**Figure 14 plants-12-04175-f014:**
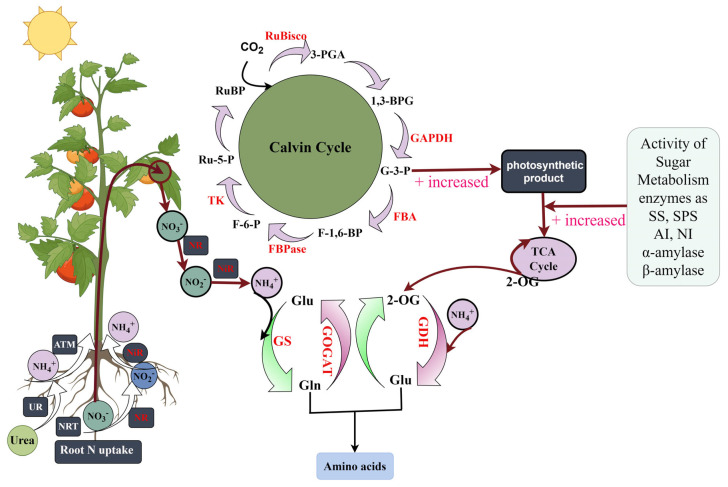
Coordination diagram showing carbon and nitrogen assimilation in tomato plants (by Figdraw https://www.figdraw.com/ accessed on 12 December 2023). The red arrow indicates the direction of substance metabolism, + indicates the promotion of the production of metabolites, the red font indicates the enzyme, and the black font indicates the metabolic substance. Organic nitrogen urea is hydrolyzed by urease (UR) to NH_4_^+^, and the root uptake and transport of inorganic nitrogen (NO_3_^−^ and NH_4_^+^) are regulated by nitrate (NRT) and ammonium (AMT) transporters through xylem. NO_3_^−^ and NH_4_^+^ are subsequently converted to amino acids by a series of nitrogen-assimilating enzymes. Similarly, the assimilated carbon during photosynthesis is converted into 2-oxoglutarate (2-OG) to further synthesize glutamic acid required for amino acid synthesis, which is used to promote tomato growth and development. Nitrate reductase (NR), nitrite reductase (NiR), glutamine synthetase (GS), glutamate synthase (GOGAT), glutamate dehydrogenase (GDH), ribulose-1,5-bisphosphate carboxylase/oxygenase (Rubisco), glyceraldehyde-3-phosphate dehydrogenase (GAPDH), fructose-1,6-bisphosphate aldolase (FBA), fructose-1,6-bisphosphate phosphatase (FBPase), transketolase (TK), ribulose 1,5-bisphosphate (RuBP), glyceric 3-phosphate (3-PGA), glyceric 1,3-bisphosphate (1,3-BPG), glyceraldehyde 3-phosphate (G-3-P), fructose 1,6-bisphosphate (F-1,6-BP), fructose 6-phosphate (F-6-P), ribulose 5-phosphate (Ru-5-P), glutamine (Gln), glutamic acid (Glu).

**Table 1 plants-12-04175-t001:** Ratios of the different forms of nitrogen and concentrations of macroelements in the nutrient solutions.

Treatments	Nitrogen Form and Ratio	Elements Concentration (mmol·L^−1^)						
		NO_3_^−^-N	NH_4_^+^-N	(CO(NH_2_)_2_)	P	K	Ca	Mg
CK	-	0	0	0	1	6	5	2
T1	NO^3−^-N (NN)	15	0	0	1	6	5	2
T2	NH_4_^+^-N (AN)	0	15	0	1	6	5	2
T3	Urea (UN)	0	0	15	1	6	5	2
T4	75%NN:25%AN	11.25	3.75	0	1	6	5	2
T5	50%NN:50%AN	7.5	7.5	0	1	6	5	2
T6	25%NN:75%AN	3.75	11.25	0	1	6	5	2
T7	25%AN:75%UN	0	3.75	11.25	1	6	5	2
T8	50%AN:50%UN	0	7.5	7.5	1	6	5	2
T9	75%AN:25%UN	0	11.25	3.75	1	6	5	2
T10	75%NN:25%UN	11.25	0	3.75	1	6	5	2
T11	50%NN:50%UN	7.5	0	7.5	1	6	5	2
T12	25%NN:75%UN	3.75	0	11.25	1	6	5	2

NN, nitrate nitrogen; AN, ammonium nitrogen; and UN, amide nitrogen.

**Table 2 plants-12-04175-t002:** Primers used for the qRT-PCR.

Gene Name	Accession Number	Forward Primer (5′-3′)	Reverse Primer (5′-3′)
*SlRbcL*	NC_007898.3	CTTTCCAAGGTCCGCCTCAT	AAGTCCACCGCGAAGACATT
*SlRbcS3*	NM_001347911.1	GCTTCTTCAGTAATGTCCTCAGC	TCCAAGCAAGGAACCCATCC
*SlFBPase*	NM_001328673.1	GGTCCAGATCAGCAATGCCT	CTCCCTGGCTGACAAACACT
*SlFBA*	NM_001321372.1	GAAGAGGAAGCCACCGTCAA	GAAGAGCACGTCCGAAGGAA
*SlTK*	XM_004248512.4	CTGTCAAGGCTGCTGAGGAA	CCCGTCAACCCCAATAGCTT
*SlGAPDH*	NM_001247874.2	AGCCACTCAGAAGACCGTTG	AGGTCAACCACGGACACATC
*SlAI*	NM_001246913.2	AACCCGCTATCTACCCGTCT	TCGGGCTTGATCCACTTACG
*SlNI*	XM_004249939.3	GCGTATGGGAAGTCCTCTGG	TACGGCGGTCTATCATGCAC
*SlSPS*	NM_001246991.2	AAAACGCCGTCAAGAACGTG	GCAATCGGCCTCTGGTACTT
*Sl* *α* *-Amylase*	XM_004238109.4	AGGCGGATGGTACAACTCTC	GCAACCGATTTGATCCCGTG
*Sl* *β* *-Amylase*	NM_001247627.2	GAAGGAAGGTGGTGGATGGG	TTGGGCGATGGGAAGGTAAC
*SlSS*	NM_001247875.1	GGTACGCCAAGAATCCACGACTAAG	CTTCTTCATCTCTGCCTGCTCTTCC
*SlNR*	Solyc11g013810.3.1	GCAACTTCCCTCCTTCATCCAAC	TCGTCATCGTCATCCTCGTCTTC
*SlNiR*	Solyc10g050890.2.1	CCGCAGAAACAGGAAGGATACAG	TGAACCATACTCATCAGCCAAACG
*SlGS1*	Solyc11g011380.2.1	GCGTCGTCTCACTGGAAAGC	TGCCTGCCTTCTCTGTGTCTC
*SlGS2*	Solyc01g080280.3.1	TACTGGACAAGCACCTGGAGAAG	AGATGTTGTTACCACCACGGAAAG
*SlGOGAT*	Solyc03g063560.3.1	GTTATGCCGCCACTAATAGGAGAAG	ATGTCATCCAAGTCAGCAACCTTAG
*SlGDH*	Solyc10g078550.4.1	GAAGACAGCGGTCGCCAATATAC	TCCAACTCAGAGATACTCAGGTCAC
*Actin*	Solyc11g005330	TGTCCCTATTTACGAGGGTTATGC	CAGTTAAATCACGACCAGCAAGAT

## Data Availability

Data are available from the corresponding author.

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
