# Peer review of "Effects of Different Forms and Proportions of Nitrogen on the Growth, Photosynthetic Characteristics, and Carbon and Nitrogen Metabolism in Tomato"

_plants, 2023, doi:10.3390/plants12244175_

Round 1
Reviewer 1 Report
Comments and Suggestions for Authors
The following points need to be addressed in the revised form:
1- The abstract should be rewritten; it should include the most important results and should summarize the experiment done.
2- References should be provided for the methods used.
3- Figures 3 and 4 should be revised for their numbering and their appearance in the text
4- Language and grammar need minor revision
5- In the materials section:
The sources of N different form should be provided
Reference to the recommended doses of fertilizer for tomato should be provided
Comments on the Quality of English Language
Minor revision is requiered
Reviewer 2 Report
Comments and Suggestions for Authors
Comments to the Author
1. This paper requires extensive language and formatting improvements.
2. Objective should be elaborated on in the introduction section.
3. References need to be rechecked and formatted as per the journal's guidelines.
4. The discussion section needs significant improvement. Currently, it is few diffuse, and provides no real insights.
5. Please add these references
Synergistic dose permutation of isolated alkaloid and sterol for anticancer effect on young swiss albino mice https://doi.org/10.2147/DDDT.S322769
Isolation and evaluation of anticancer efficacy of stigmasterol in a mouse model of DMBA-induced skin carcinoma DOI https://doi.org/10.2147/DDDT.S83514
require major revisions before publication.
Comments on the Quality of English Languagemajor revisions
Reviewer 3 Report
Comments and Suggestions for Authors
Dear Authors,
I believe the introduction is very well articulated and written, addressing the state of the art appropriately and outlining the objectives that this work aims to address correctly. I believe the article has substantial experimental content, and the results are presented and discussed correctly. However, it is true that studies of this kind are numerous in the literature with other types of plants, making the study itself less original. I don't see the authors referencing similar studies conducted with tomatoes and discussing their results in comparison with those available in the literature. In my opinion, the authors should improve on this aspect.
Majors:
-What nitrogen source was used for the control (CK)? Please specify the concentration and the type exactly.
-The materials and methods seem to be described in a very detailed, specific, and accurate manner. I just have one question; on many occasions, the authors mention: “three tomato plants with uniform growth were selected from each treatment” I think choosing only 3 plants for each of the treatments is too low a number. Why didn't the authors select a larger number to increase reproducibility?
-I don't see it appropriate to include Figure 2, as differences are very difficult to discern at a glance, and moreover, they are unique examples for each condition. Additionally, the results are quantified in Table 3; I advise the authors to remove it, as it contributes little scientifically. I also advise that the results in Table 3 be presented as column graphs to facilitate comprehension.
-Change the order of Figures 3 and 4, as in the text, Figure 4 is mentioned before Figure 3.
- I believe the discussion is well-conducted, effectively comparing the obtained results with those of other authors in a very appropriate manner regarding other plants. However, concerning other studies that have also investigated the effect of different nitrogen fertilizers on tomatoes, this point needs to be substantially corrected.
-L733-L752: I advise the authors to replace this paragraph used as a summary with another that plays a role in presenting the main conclusions of the study.
Minors:
L551: “Some researchers believe that the accumulation of ammonium nitrogen in plant leaves may lead to the uncoupling of electron transport in chloroplasts to form phos phorylation” I'm sorry, I don't understand this sentence. Could you please rephrase it?
L674: “ammonium is the primary storage and transportation form of ammonium” I don't see the point of this sentence. Could you provide a better explanation of what you mean?
Round 2
Reviewer 2 Report
Comments and Suggestions for Authors
accept
Comments on the Quality of English Languageaccept
Author Response
Thank you for your comments and suggestions on our manuscripts. Again ,my heartfelt thanks.
Reviewer 3 Report
Comments and Suggestions for Authors
Dear Authors,
I believe the authors have responded appropriately to all my comments and suggestions, and I accept the paper in its current version.
Author Response

(The authors gave the same response as above.)
